# Dynamic Changes in Vitamin E Biosynthesis during Germination in Brown Rice (*Oryza sativa* L.)

**DOI:** 10.3390/foods11203200

**Published:** 2022-10-13

**Authors:** Leilei Kong, Yingxin Lin, Jiayan Liang, Xiaodan Hu, Umair Ashraf, Xinbo Guo, Song Bai

**Affiliations:** 1Rice Research Institute, Guangdong Academy of Agricultural Sciences/Guangdong Key Laboratory of New Technology in Rice Breeding/Guangdong Rice Engineering Laboratory, Guangzhou 510640, China; 2School of Food Science and Engineering, South China University of Technology, Guangzhou 510640, China; 3Department of Botany, Division of Science and Technology, University of Education, Lahore 54770, Punjab, Pakistan

**Keywords:** brown rice, germination, vitamin E, biosynthesis

## Abstract

The present study investigated the dynamic changes in vitamin E and gene expression within its biosynthetic pathway during three germination periods of four brown rice cultivars with different seed coat colors. The results reveal that the vitamin E content increased during the germination process of all brown rice cultivars. Moreover, the content of α-tocopherol, α-tocotrienol, and β-tocopherol significantly increased at the later stage of germination. The expression levels of *DXS1* and *γ-TMT* genes in all cultivars were significantly increased, whilst the *HGGT* gene expression levels of G6 and XY cultivars increased significantly at the later stage of brown rice germination. In addition, the expression levels of *MPBQ/MT2* in G1 and G6 cultivars, and *TC* expression levels in G2 and G6 cultivars were substantially increased at the later stage of germination. Overall, the up-regulation of *MPBQ/MT2*, *γ-TMT,* and *TC* genes doubled the content of α-tocopherol, α-tocotrienol, and β-tocopherol, and the total vitamin E content of brown rice was at its highest at 96HAT. The utilization of the germination period can effectively improve the nutritional value of brown rice, which can be used for the development and utilization of brown rice for healthy rice products.

## 1. Introduction

Brown rice (*Oryza sativa* L.) is rich in nutrients with excellent edible value [1,2]. The important nutrients in rice, such as vitamins, minerals, amino acids, and dietary fiber, are concentrated in the outer tissue, while the endosperm part, white rice, is mainly composed of 90% starch and 9% protein [3,4]. In general, brown rice is hard and loose, with poor viscoelasticity, heavy bran flavor and a coarse taste, consequently, making it difficult to cook. Therefore, it is difficult for most consumers to accept brown rice as a staple food. On the other hand, seed germination is a physiologically active process [5], and there are few reports on the changes in the concentrations of vitamins during the seed germination phase. For example, Esa et al. found that the vitamin E content of brown rice increased significantly after germination [6]; likewise, Spanier et al. reported a nearly four-fold increase in the vitamin E content of brown rice after germination [7].

Insulin activity, glycolysis, glycogenesis, lipid synthesis, and lipid absorption can be improved in brown rice during germination [8]. Many researchers have proved that germinated brown rice has pharmacological effects; i.e., it helps in lowering blood fat and blood pressure, prevents diabetes and assists in the treatment of the condition and its complications, reduces the incidence of cardiovascular diseases, improves memory, and prevents senile dementia [9,10,11,12,13].

Germinated brown rice has many antioxidant components, including vitamin E, which can inhibit tumor cell proliferation and reduce the incidence of cardiovascular diseases [14]. Vitamin E, as an essential nutrient element, is mainly distributed in the leaves and the fruits of plants in nature. In higher plants, vitamin E can be synthesized by photosynthesis and is divided into tocopherols and tocotrienols, which are further divided into α-, β-, γ-, and δ- configurations [15]. The biosynthetic pathway of vitamin E in rice is shown in Figure 1 [16,17].

The biosynthesis of vitamin E mainly uses the shikimate metabolic (SK) pathway product, homogentisic acid (HGA) as a hydrophilic head, and uses the methylerythritol phosphate (MEP) pathway product phytyldiphosphate (PDP) or geranylgeranyl-diphosphate (GGDP) as a hydrophobic tail of tocopherol or tocotrienol [18,19]. At least seven enzymes are known to be directly involved in vitamin E synthesis. Firstly, 1-deoxy-D-xylulose-5-phosphate synthase (DXS) is the first speed limit enzyme of the MEP pathway. It can synthesize 1-deoxy-D-xylulose-5-phosphate (DXP) from the substrate D-glyceraldehyde-3-phosphate (D-GAP) and pyruvate with the aid of thiamine pyrophosphate (TPP), thus forming GGDP and PDP [20,21]. Moreover, 4-hydroxyphenyl pyruvate dioxygenase (HPPD) is the first key enzyme in the synthesis of vitamin E, which catalyzes the conversion of 4-hydroxyphenylpyruvate (HPP) into homogentisic acid (HGA) in the cytoplasm [22]. Next, homogentisic acid phytyltransferase (HPT) catalyzes HGA and PDP to form 2-methyl-6-phytylquinol (MPBQ). Then, MPBQ forms 2, 3-dimethyl-5-phytylquinol (DMPBQ) under the action of MPBQ methyltransferase (MPBQ-MT). γ-tocopherol and δ-tocopherol can be directly generated from MPBQ and MPBQ-MT under tocopherol cyclase (TC) catalysis, respectively. Then, α-tocopherol and β-tocopherol can be formed under the action of γ -tocopherol methyltransferase (γ-TMT) from γ-tocopherol and δ-tocopherol, respectively [22,23,24,25,26]. On the other hand, homogentisate geranylgeranyltransferase (HGGT) catalyzes the formation of 6-geranylgeranyl-2-methylbenzene-1,4-diol (GGBM) from HGA and GGDP and then produces 6-geranylgeranyl-2,3-dimethylbenzene-1,4-diol (GGDMB). Next, δ-tocotrienol and γ-tocotrienol are directly converted from GGDMB and GGBM by TC. β-tocotrienol and α-tocotrienol can be directly generated from δ-tocotrienol and γ-tocotrienol by γ-TMT, respectively [22,23,24,25,26].

All the homologous genes related to vitamin E synthesis in rice were identified in the Rice Genome Annotation Project [27] e.g., Osγ-TMT (LOC_Os02g47310), OsHPPD (LOC_Os02g07160), OsTC (LOC_Os02g17650) on chromosome 2, OsHPT (LOC_Os06g44840), and OsHGGT (LOC_Os06g43880) on chromosome 6, as well as, OsMPBQ/MT2 (LOC_Os12g42090) on chromosome 12. However, there are few studies on the dynamic changes in the key gene expression of vitamin E metabolic pathways during brown rice germination. Therefore, this study can provide a theoretical basis for the research into the nutritional quality and the processing quality of germinated brown rice.

## 2. Materials and Methods

### 2.1. Chemicals and Reagents

Vitamin E standards for high-performance liquid chromatography (HPLC) were purchased from Sigma Aldrich (St. Louis, MO, USA). Methanol, hexane, isopropyl alcohol, acetic acid, and ammonium acetate of HPLC grade were purchased from ANPEL Scientific instrument Co., Ltd. (Shanghai, China). Other chemicals and reagents used were of analytical grade.

The pot experiment was conducted in an artificial climate chamber (RDN-1000B-4, Ningbo southeast Instrument Co., Ltd., Ningbo, China) in the Rice Research Institute, Guangdong Academy of Agricultural Sciences, Guangzhou, China. The conditions inside the climate chamber were set at a temperature of 30 °C, in a dark environment, with a relative air humidity of 80%.

### 2.2. Germinated Brown Rice (GBR) Preparation

Treatment of germination: Firstly, impure, abortive, and damaged grains in the brown rice were removed. Secondly, the brown rice was rinsed in a plastic basin with tap water 5 times, the water was drained, the rice was disinfected by soaking in a 1% sodium hypochlorite solution for 15 min, this solution was drained, and the rice was then washed in ultra-pure water 5 times. Thirdly, the rice was soaked in an artificial climate chamber at 30 °C for 18 h, the liquid was poured out, and the soaked brown rice was placed on a sterilized tray with two layers of filter paper covering it to prevent moisture loss. The brown rice was germinated in an artificial climate chamber set at 30 °C, in a dark environment, with a relative air humidity of 80%, and it was sprayed with ultra-pure water every 6 h to keep it moist. Seeds of four different colored rice varieties i.e., Guanghei1 (G1), Guichao2 (G2), Guanghong6 (G6), and Xiangyaxiangzhan (XY), with dark brown, white, red, and white seed covers, respectively, were placed in Petri dishes. The sampling was conducted over three time intervals i.e., 2 h after seed soaking treatment (2HAT), 48 h after seed soaking treatment (48HAT), and 96 h after seed soaking treatment (96HAT). The morphological appearances of different brown rice varieties at different stages of germination are shown in Figure 2. Before each sampling, the seed surface was dried with paper towels and frozen in liquid N_2_ and stored at -80 °C for further analyses.

### 2.3. Extraction and Determination of Vitamin E

The extraction and determination of vitamin E content were conducted using the method described by Xiang et al. and Xie et al. [28,29]. Vitamin E extractions were re-dissolved with 1% isopropanol/hexane. Vitamin E presence was determined with a Waters HPLC system at 290 nm excitation wavelength and 330 nm emission wavelength [30]. Hexanes/isopropyl alcohol/acetic acid (99.05:0.85:0.1, *v/v/v*) were used as a mobile phase with 1.0 mL min^−^^1^ through a silica column (Zorbax RX-SIL 4.6 × 250 mm, 5 µm, Agilent Technologies Inc., Santa Clara, CA, USA). The A’ phase contains 0.05 mol L^−^^1^ ammonium acetate with 0.1% butylated hydroxytoluene (*w/v*) in 97% methanol—water solution and the B0 phase contains 0.1% BHT (*w/v*) in methyl tert-butyl ether. The vitamin E isomer content was quantitated by contrasting with eight external standards based on retention time. Four tocopherol standards were obtained from Wako Pure Chemical Industries (Tokyo, Japan), while four tocotrienol standards were acquired from Chromadex, Ltd. (Irvine, CA, USA). Each vitamin E isomer content was expressed as µg/g DW, and the total vitamin E isomer content was the sum of both tocopherols and tocotrienols from each sample. Data are reported as mean ± SD (*n* = 3).

### 2.4. Real-Time Quantitative Polymerase Chain Reaction (RT-qPCR) Analysis

The extraction of RNA was accomplished using an HP Plant RNA Kit (Omega, Norcross, GA, USA). Reverse transcription of RNA was conducted using a Fast King RT Supermix Kit with g DNase (Tiangen, Beijing, China) according to the manufacturer’s instructions. The RT-qPCR was conducted using a Super Real Pre Mix Plus (SYBR Green) kit (Tiangen, Beijing, China) according to the manufacturer’s instructions, together with a Light Cycler^®^ 480 Real-Time PCR System (F. Hoffmann-La Roche Ltd., Basel, Switzerland). The nucleotide sequences of primers used in RT-qPCR are presented in Table 1. Actin depolymerizing factor (ADF) was tested as the reference gene, and the cycle threshold (Ct) values were used to calculate the expression levels using the 2^−ΔΔCt^ method. Results are reported as a mean ± SE (*n* = 3). “Primer Premier” software was used to design primers according to the known sequences of each gene and the primer design principle of fluorescence quantitative PCR (Table 1).

### 2.5. Statistical Analysis

Data were analyzed using Origin Pro 2021 (Origin Lab Corporation, Northampton, MA, USA) and Statistix 8.1 (Analytical Software, Tallahassee, FL, USA). Differences amongst means were separated by using the least significant difference (LSD) test *p* < 0.05, and the correlation analysis was conducted using the Pearson correlation coefficient.

## 3. Results

### 3.1. Vitamin E Profiles and Composition in Brown Rice after Germination

According to the HPLC analysis, five isomers of vitamin E were detected in brown rice after germination, including α-tocopherol/tocotrienol, γ-tocopherol/tocotrienol, and β-tocopherol. Both δ-tocopherol/tocotrienol and β-tocotrienol were below the detectable limit in the brown rice in this study. From the heat map and the bar graph (Figure 3 and Figure 4), it was found that for G2 and G6, with the extension of germination time, the α-tocopherol and α-tocotrienol content increased significantly, and the content of α-tocopherol and α-tocotrienol at 96HAT increased by 2.41 and 2.39 times, respectively, as compared with 2HAT.

Furthermore, the content of γ-tocopherol decreased significantly with the prolongation of the germination time of brown rice. The content of γ-tocopherol in G1, G2, G6, and XY varieties at 96HAT significantly decreased by 14.57%, 25.86%, 30.86%, and 18.85% compared with 2HAT, respectively. The content of γ-tocotrienol decreased in the G2 cultivar at 48HAT, but there was no significant change in other cultivars at the germination stage. Significant differences among varieties were noted for β-tocopherol and β-tocotrienol. Both G6 and XY varieties produced β -tocopherol during the whole germination process of brown rice, the β -tocopherol content of G6 increased with the increase in germination time of brown rice, and the β -tocopherol content at 96HAT was significantly higher than that at 2HAT.

### 3.2. Gene Expression Profiles Related with Vitamin E Biosynthesis

The expression levels of genes involved in vitamin E synthesis are shown in the heat map and the bar graph (Figure 5 and Figure 6, respectively). The expression of *DXS1* in all varieties showed a gradual increasing trend. The expression level of *DXS1* in G2 was the highest at 48HAT, which was 9.50 times higher when compared with 2HAT. The gene expression level of *DXS1* in G2 initially increased and then decreased and was found to be significantly higher than that at 2HAT. Moreover, the gene expression level of *DXS1* in G1 increased steadily, and no significant difference was found at 48HAT when compared with 2HAT; however, it reached its peak at 96HAT, with a 6.42 times higher expression level when compared with 2HAT. The gene expression level of *DXS1* in G6 gradually decreased and then gradually rose, peaking at 96HAT, with its expression being the same as G1. The gene expression level of *DXS1* in XY at 48HAT and 96HAT was significantly higher than that at 2HAT but remained statistically similar (*p* > 0.05).

The expression levels of the *HGGT* gene in all cultivars decreased first and then increased, with a significant increase at 96HAT when compared with 48HAT. The *HGGT* gene expression levels of G6 and XY increased significantly at 96HAT when compared with 2HAT and 48HAT. The *HGGT* gene expression levels of G6 and XY at 96HAT increased 1.45 times and 0.95 times when compared with 2HAT, respectively. Furthermore, the gene expression level of *HPPT* firstly decreased and then increased with the extension of germination time; in contrast, the gene expression of *HPPD* in G2 gradually decreased with the extension of germination time. However, the gene expression of *HPPD* in brown rice was significantly lower than that before germination.

The gene expression level of *HPT* was significantly different among the varieties. The expression level of XY at 96HAT was 2.87 times higher than that at 2HAT, while the gene expression level of *HPT* in other varieties decreased with the prolonged germination time of brown rice. Except for G2, the gene expression level of *MPBQ/MT2* was significantly increased after the germination of brown rice. The gene expression levels of G1, G6, and XY at 96HAT were 8.55, 9.70, and 2.01 times higher than those at 2HAT, respectively. Moreover, the gene expression level of *TC* in G2 and G6 at 96HAT was 1.55 times and 8.02 times higher than that at 2HAT but decreased and/or had no significant difference in other varieties. The gene expression level of *γ-TMT* in the four varieties increased significantly with the increase in germination time, and the expression level at 96HAT increased by 3.25, 66.00, 5.10, and 21.71 times, respectively, when compared with 2HAT.

### 3.3. Correlations among Vitamin E and Genes Involved in Its Biosynthesis

During the germination of brown rice, significant positive correlations between the total vitamin E content and the content of α-tocopherol, α-tocotrienol, and γ-tocotrienol were noted (Figure 7). In addition, a significant positive correlation was found between the gene expression levels of *MPBQ/MT2* and *TC*, whereas a significant negative correlation was found between the gene expression levels of *HPPD*. There were significant positive correlations between α-tocopherol and α-tocotrienol, and between the α-tocotrienol and γ-tocopherol. On the other hand, the contents of α-tocopherol and α-tocotrienol were significantly and negatively correlated with the content of γ-tocopherol. At the same time, the content of α-tocopherol showed a significant positive correlation with the gene expression levels of *DXS1*, *TC,* and *γ-TMT*; a significant negative correlation with the gene expression levels of *MPBQ/MT2*; and a significant negative correlation with the gene expression levels of *HPPD*. 

In addition, the content of α-tocotrienol had a significant positive correlation with the gene expression of *HGGT* and *TC*. The content of γ-tocopherol also had a significant positive correlation with the gene expression levels of *HPPD*, and significant negative correlations with the gene expression levels of *DXS1*, *γ-TMT,* and *TC*. Moreover, γ-tocotrienol had a significant positive correlation with *TC* gene expression.

## 4. Discussion

Rice is one of the most important food crops in the world, and people mainly consume processed/polished rice [31,32]. The nutrients of brown rice are far superior to those of white rice. At present, people adopt various ways to improve the nutritional value of rice, with germinated brown rice being the most common [33,34,35]. Brown rice germination promotes the increase in a variety of bioactive compounds, including gamma-aminobutyric acid (GABA) and antioxidants [36,37,38]. However, there are few studies on the anabolism of vitamin E during the germination of brown rice. In this study, the total vitamin E content of brown rice was increased after germination, mainly with the increase in α-tocopherol and α-tocotrienol, which is consistent with the study conducted by Kim et al. [39]. The change in vitamin E content was significantly correlated with the rice variety and the germination time, which is consistent with the study conducted by Yodpitak et al. [40]. Among the eight homologues, γ-tocotrienol was present in the highest amounts in all samples, but there was no significant change during germination, whereas the α-tocopherol content of brown rice was significantly increased during germination. These results are a little different from those of Kim et al., where the α-tocopherol and γ-tocotrienol content of brown rice increased after germination [41]. The content of β-tocopherol was only detected in G1 and G2 at 96HAT, which may be due to the low and slightly decreased expression level of *TC* in the early stage of germination and the significantly increased expression of *TC* and *TMT* in the late stage of germination at 96HAT, promoting the transformation of MPBQ into β-tocopherol. G6 also had the highest level of β-tocopherol at 96HAT, possibly because the *TC* gene expression level of G6 was the highest among all four varieties at 96HAT and had increased by 8.02 times when compared with 2HAT, which caused the formation of β -tocopherol directly via the shunt catalysis of MPBQ through TC and TMT. 

GBR has a high nutritional value. The increase in nutrient content during germination of brown rice is due to the action of a series of key enzymes. In particular, the up-regulation of *MPBQ/MT2*, *γ-TMT,* and *TC* genes doubled the content of α-tocopherol, α-tocotrienol, and β-tocopherol, and the total vitamin E content of brown rice was the highest at 96 h after germination.

Overall, brown rice germination can promote the production of α-tocopherol and α-tocotrienol, mainly due to the increase in *γ-TMT* and *TC* gene expression, which leads to more conversion of γ-tocopherol and γ-tocotrienol into α-tocopherol and α-tocotrienol. On the other hand, it can be seen that the increase in *MPBQ/MT2* gene expression promoted the transformation of MPBQ to DMPBQ. In the late germination stage of brown rice, the increase in *TC* gene expression and the effect of *γ-TMT* led to an increase in α-tocopherol content and a decrease in γ-tocopherol content. The possible reason is that the increase in *γ-TMT* gene expression is far greater than the increase in *TC* gene expression, so more γ-tocopherol produced by DMPBQ is transformed into α-tocopherol. At the same time, *HGGT* may play a more important role in the production of γ-tocotrienol in brown rice, and the increase in *γ-TMT* gene expression at the late germination stage promotes the conversion of γ-tocotrienol to α -tocotrienol. Finally, due to a series of key enzymes, especially MPBQ/MT2, γ-TMT, and TC, the content of α-tocopherol, α-tocotrienol, and β-tocopherol in brown rice during germination increase exponentially, which ultimately promotes an increase in total vitamin E content.

## 5. Conclusions

Germinated brown rice has a high nutritional value and is promoted as a raw material of good food for the benefit of human health. During the germination process of brown rice, the total vitamin E content of colored brown rice (G1 and G6) increased significantly with the extension of germination time, while the total vitamin E content of G2 and XY firstly decreased and then increased with an increase in germination time. Overall, the utilization of a germination period can effectively improve the nutritional value of brown rice, which can be used for the development and utilization of brown rice for healthy rice products.

## Figures and Tables

**Figure 1 foods-11-03200-f001:**
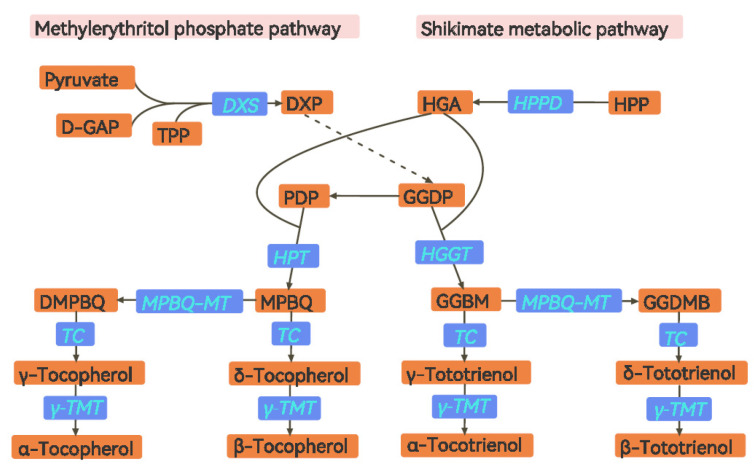
Vitamin E biosynthesis pathway in rice. D-glyceraldehyde-3-phosphate (D-GAP), thiamine pyrophosphate (TPP), 1-deoxy-d-xylulose-5-phosphate (DXP), DXP synthase (DXS), 4-hydroxyphenylpyruvate (HPP), 4-hydroxy-phenylpyruvate dioxygenase (HPPD), homogentisic acid (HGA), geranylgeranyl diphosphate (GGDP), phytyldiphosphate (PDP), homogentisate phytyltransferase (HPT), homogentisate geranylgeranyltransferase (HGGT), 2-methyl-6-phytylquinol (MPBQ), 2,3-dimethyl-5-phytylquinol (DMPBQ), MPBQ methyltransferase (MPBQ-MT), tocopherol cyclase (TC), γ-tocopherol methyltransferase (γ-TMT), 6-geranylgeranyl-2-methylbenzene-1,4-diol (GGBM), 6-geranylgeranyl-2,3-dimethylbenzene-1,4-diol (GGDMB). Same below, as shown in Table A1.

**Figure 2 foods-11-03200-f002:**
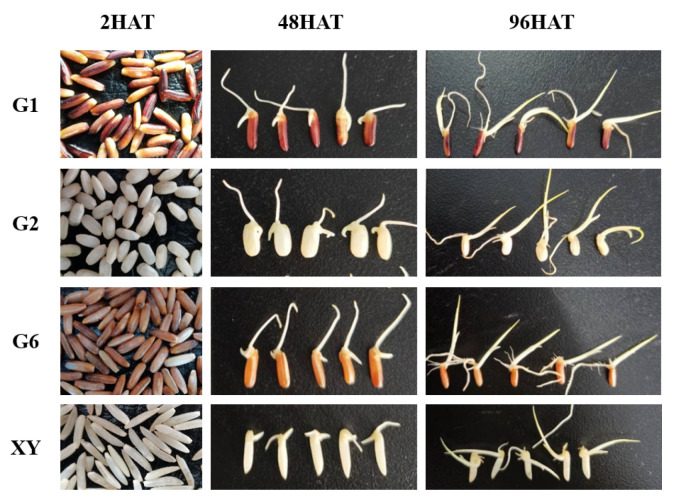
Morphological diagram of brown rice at different stages of germination. Treatments: 2 h after seed soaking treatment (2HAT), 48 h after seed soaking treatment (48HAT), and 96 h after seed soaking treatment (96HAT). Varieties: Guanghei1 (G1), Guichao2 (G2), Guanghong6 (G6), Xiangyaxiangzhan (XY).

**Figure 3 foods-11-03200-f003:**
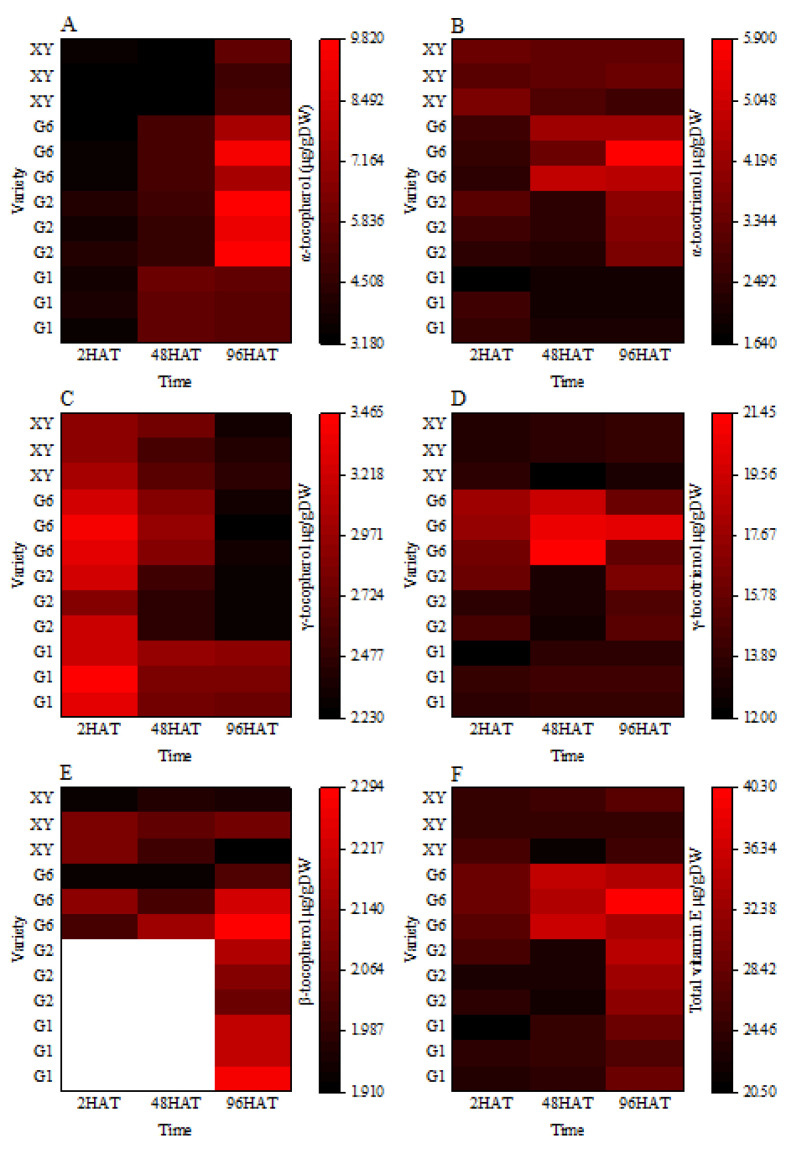
Heat map of the dynamic changes in vitamin E isomer content in brown rice during germination. (**A**): The α-tocopherol content. (**B**): The α-tocotrienol content. (**C**): The γ-tocopherol content. (**D**): The γ- tocotrienol content. (**E**): The β-tocopherol content. (**F**): The total vitamin E content. Treatments: 2 h after seed soaking treatment (2HAT), 48 h after seed soaking treatment (48HAT), 96 h after seed soaking treatment (96HAT). Varieties: Guanghei1 (G1), Guichao2 (G2), Guanghong6 (G6), Xiangyaxiangzhan (XY). The white section in the heat map was not detected.

**Figure 4 foods-11-03200-f004:**
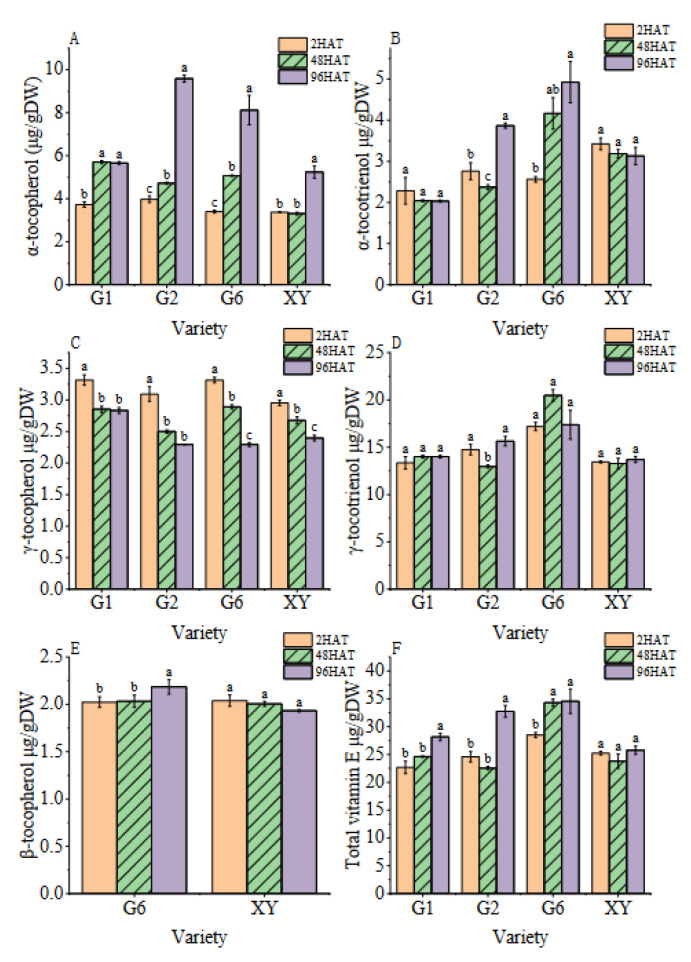
Bar chart of dynamic changes in vitamin E isomer content in brown rice during germination. (**A**): The α-tocopherol content. (**B**): The α-tocotrienol content. (**C**): The γ-tocopherol content. (**D**): The γ- tocotrienol content. (**E**): The β-tocopherol content. (**F**): The total vitamin E content. Treatments: 2 h after seed soaking treatment (2HAT), 48 h after seed soaking treatment (48HAT), 96 h after seed soaking treatment (96HAT). Varieties: Guanghei1 (G1), Guichao2 (G2), Guanghong6 (G6), Xiangyaxiangzhan (XY). Bars with different letters differ significantly at *p* < 0.05.

**Figure 5 foods-11-03200-f005:**
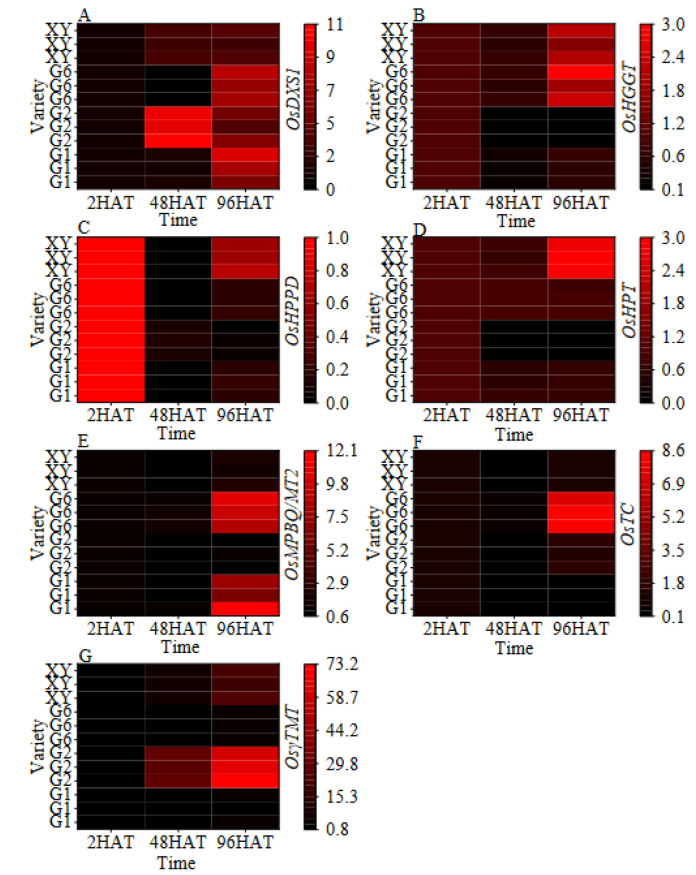
Heat map of the dynamic changes in vitamin E synthetic gene expression during germination of brown rice. (**A**): The *OsDXS1* gene expression. (**B**): The *OsHGGT* gene expression. (**C**): The *OsHPPD* gene expression. (**D**): The *OsHPT* gene expression. (**E**): The *OsMPBQ/MT2* gene expression. (F): The *OsTC* gene expression. (**G**): The *Os**γ-TMT* gene expression. Treatments: 2 h after seed soaking treatment (2HAT), 48 h after seed soaking treatment (48HAT), 96 h after seed soaking treatment (96HAT). Varieties: Guanghei1 (G1), Guichao2 (G2), Guanghong6 (G6), Xiangyaxiangzhan (XY).

**Figure 6 foods-11-03200-f006:**
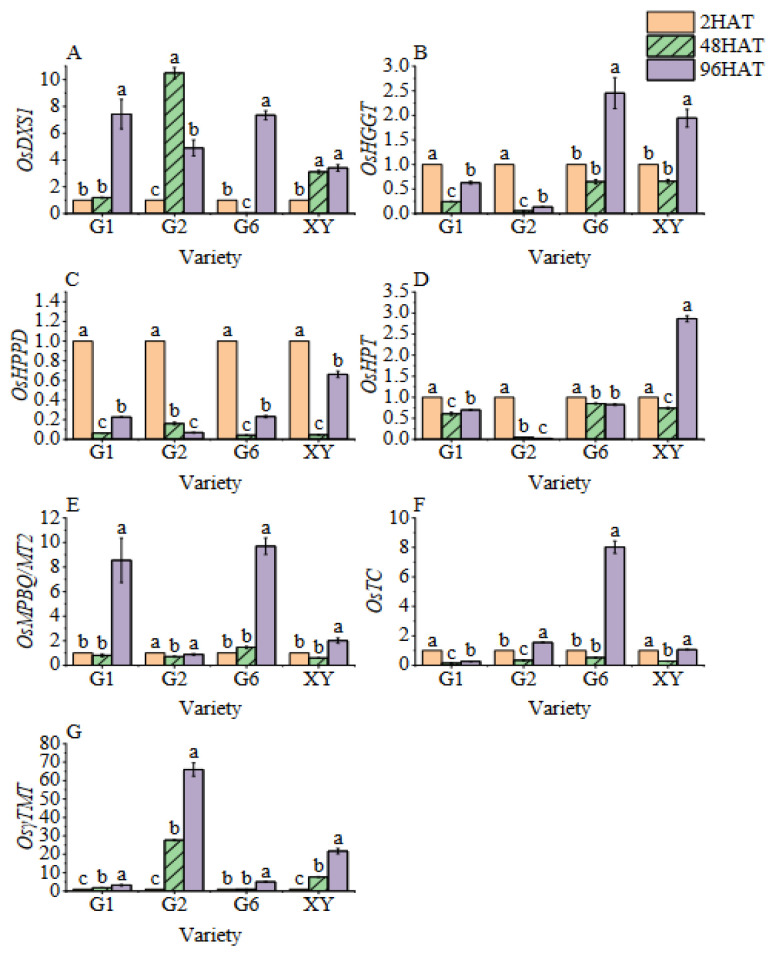
Bar chart of the dynamic changes in vitamin E synthetic gene expression during germination of brown rice. (**A**): The *OsDXS1* gene expression. (**B**): The *OsHGGT* gene expression. (**C**): The *OsHPPD* gene expression. (**D**): The *OsHPT* gene expression. (**E**): The *OsMPBQ/MT2* gene expression. (**F**): The *OsTC* gene expression. (**G**): The *Os**γ-TMT* gene expression. Treatments: 2 h after seed soaking treatment (2HAT), 48 h after seed soaking treatment (48HAT), 96 h after seed soaking treatment (96HAT). Varieties: Guanghei1 (G1), Guichao2 (G2), Guanghong6 (G6), Xiangyaxiangzhan (XY). Bars with different letters differ significantly at *p* < 0.05.

**Figure 7 foods-11-03200-f007:**
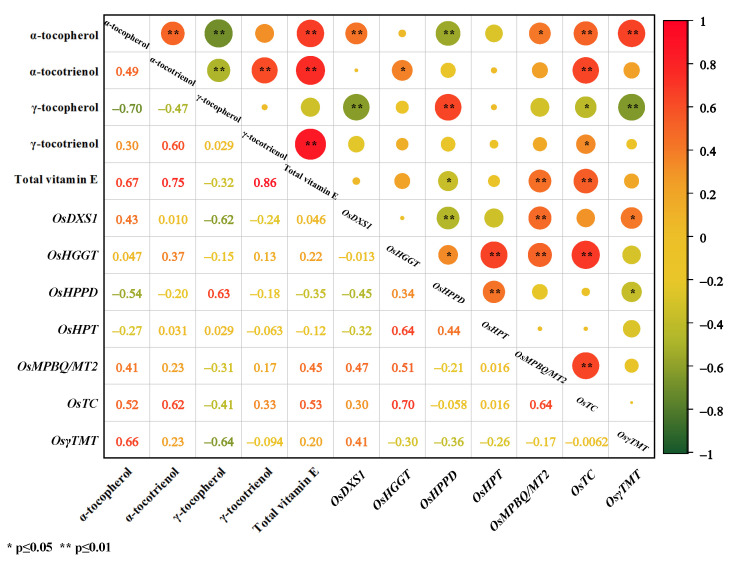
Correlations among vitamin E and the genes of vitamin E biosynthesis during brown rice germination. The strength of the correlation between two variables is represented by the color of the circle at the intersection of those variables. Colors range from bright orange (strong positive correlation; i.e., r^2^ = 1.0) to bright green (strong negative correlation; i.e., r^2^ = −1.0).

**Table 1 foods-11-03200-t001:** The nucleotide sequences primers of vitamin E synthesis-related enzymes.

Gene Name	Primers
*Osγ-TMT* (LOC_Os02g47310)	F 5′-CCAGACTGGTGCTCTCCTTC-3′
	R 5′-CATCAGAGGCATCACCATTG-3′
*OsHPPD* (LOC_Os02g07160)	F 5′-AGGAGACAGGCCAACCTTTT-3′
	R 5′-TGAACTGTAGGGGCTTGCTT-3′
*OsTC* (LOC_Os02g17650)	F 5′-ATGTCTTCTCAGGCGCATCT-3′
	R 5′-GTGCCTGGTTCTTTTGTGGT-3′
*OsHPT* (LOC_Os06g44840)	F 5′-GTCCGATGTGTCTCCCTTGT-3′
	R 5′-TCCCCAGATGCTAATGGAAG-3′
*OsHGGT* (LOC_Os06g43880)	F 5′-AACAAAGTCGGTGGTTTTCG-3′
	R 5′-GATGATGCTCCAGCCAAAAT-3′
*OsMPBQ/MT2* (LOC_Os12g42090)	F 5′-AGTTCTTATGAGCTTAATCAAGGT-3′
	R 5′-TTTCTGTCAGTTCTGTATTTACTTCTGTTG-3′
*OsDXS1* (NM_001062059)	F 5′-ACCAAACGCTCATCAGGAGG-3′
	R 5′- GTGGTCGATGTACCTGTCGG -3′
*Ubi-Q*	F 5′- ACCACTTCGACCGCCACTACT -3′
	R 5′- ACGCCTAAGCCTGCTGGTT -3′

## Data Availability

Not applicable.

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
