# Peer review of "Dynamic Changes in Vitamin E Biosynthesis during Germination in Brown Rice (Oryza sativa L.)"

_foods, 2022, doi:10.3390/foods11203200_

Round 1

Reviewer 1 Report

The authors studied dynamic changes of vitamin E content during brown rice germination and correlated it to the expression of key genes of vitamin E biosynthesis.

The topic of the present manuscript is interesting and cover a good scope, relevant to the Journal. The manuscript is well organized and written. The data reported are relevant and well discussed.

 some minor modifications and comments:

-        -   Line 57: please delete “Figure 1”

-         - Materials and methods section: did the standards used for HPLC vitamin E determination includes all the homologues? Please clarify.

-         -  Line 112: in figure 2 are reported changes of vitamin E during germination, not morphological data. Please, correct.

-         -  In the figure of Heat map of dynamic changes of vitamin E some data are lacking (the color of some bands for b-tocopherol is lacking)

-         -  Line 125: please clarify how did you identify and quantify the vitamins homologues.

-          - Line 151: Please check and correct the number of all the figures of paragraphs 3.1 and 3.2.

-          - Line 240: I think is better to use the term “germination” instead of “generation”.

-          - Line 242: please change “that” with “where”.

-          - Line 251: GRB is GBR?

-          - Lines 251-252: the sentences are not clear. Please, check and correct.

-         -  Bibliography: please check and correct the citation format.

Reviewer 2 Report

This manuscript entitled “Dynamic changes in biosynthesis of vitamin E during germination in brown rice” is an interesting and original study.

The paper is clearly presented and results are very useful. However, I have some suggestions:

1.     Review the figure captions of the manuscript. The figures must be independent and understandable by themselves. Indicate what each code is in the figure caption.

2.     Figure 3 is not related to correlations. Why does it indicate in its caption some correlations?
